# Hypertensive Stimuli Indirectly Stimulate Lymphangiogenesis through Immune Cell Secreted Factors

**DOI:** 10.3390/cells11142139

**Published:** 2022-07-07

**Authors:** Brooke K. Wilcox, Marissa R. Henley, Shobana Navaneethabalakrishnan, Karina A. Martinez, Anil Pournouri, Bethany L. Goodlett, Alexandra H. Lopez, Miranda L. Allbee, Emma J. Pickup, Kayla J. Bayless, Sanjukta Chakraborty, Brett M. Mitchell

**Affiliations:** College of Medicine, Texas A&M University, Bryan, TX 77807, USA; brookewilcox23@tamu.edu (B.K.W.); marissahenley19@tamu.edu (M.R.H.); navshobi27@exchange.tamu.edu (S.N.); kamartinez5@buffs.wtamu.edu (K.A.M.); anil.pournouri@ace.tamut.edu (A.P.); bethany.goodlett@tamu.edu (B.L.G.); alexhopelopez@gmail.com (A.H.L.); allbee2022@tamu.edu (M.L.A.); emma.j.pickup@uth.tmc.edu (E.J.P.); kaylab@tamu.edu (K.J.B.)

**Keywords:** lymphatic endothelial cells, immune cells, hypertension

## Abstract

(1) Background: Renal immune cells and lymphatic vessel (LV) density have been reported previously to be increased in multiple mouse models of hypertension (HTN). However, whether interstitial levels of HTN stimuli such as angiotensin II, salt, or asymmetric dimethylarginine have a direct or indirect effect on lymphangiogenesis is unknown. We hypothesized that these 3 HTN stimuli directly increase lymphatic endothelial cell (LEC) proliferation, LEC 3-D matrix invasion and vessel formation, and sprouting of mouse mesometrial LVs. (2) Methods: Human LECs (hLECs) and mouse LECs (mLECs) were treated with HTN stimuli while explanted mouse mesometrial LVs were treated with either the same HTN stimuli or with HTN stimuli-conditioned media. Conditioned media was prepared by treating murine splenocytes with HTN stimuli. (3) Results: HTN stimuli had no direct effect on hLEC or mLEC proliferation. Treatment of hLECs with HTN stimuli increased the number of lumen-forming structures and invasion distance (both *p* < 0.05) in the 3-D matrix but decreased the average lumen diameter and the number of cells per invading structure (both *p* < 0.05). Conditioned media from HTN-stimuli-treated splenocytes significantly attenuated the decrease in sprout number (aside from salt) and sprout length of mouse mesometrial LVs that is found in the HTN stimuli alone. (4) Conclusions: These data indicate that HTN stimuli indirectly prevent a decrease in lymphangiogenesis through secreted factors from HTN-stimuli-treated immune cells.

## 1. Introduction

Hypertension (HTN), defined as having a blood pressure of 130/80 mmHg or higher, affects nearly half of all American adults and is a leading risk factor for cardiovascular disease, the most common cause of death in the United States [1]. HTN increases the flow of blood, and sustained high-pressure conditions can lead to end organ damage, organ failure, and death [2]. Lowering blood pressure by just 10 mmHg reduces the risk of death and improves patient health outcomes; however, many patients struggle to regulate their blood pressure [3].

Lymphatic vessels (LVs) drain extravasated fluid, immune cells, and proteins from the interstitial fluid and transport them to draining lymph nodes [4]. LVs contain a thin layer of lymphatic endothelial cells (LECs), which act as a pathway for lymph and immune cells and play a critical role in the function of the immune system [5]. LECs also aid in lymphatic draining under conditions of high interstitial pressure [5]. Under acute inflammatory conditions, the body’s first response is to dilate lymphatic capillaries. If inflammation persists, the formation of new LVs occurs to improve the trafficking of infiltrating activated immune cells [6]. Inflammation-associated lymphangiogenesis has been identified in numerous inflammatory disease conditions in the heart, kidney, skin, lungs, and intestines [7]. We demonstrated previously that compensatory lymphangiogenesis occurs in the kidneys of mice with HTN; however, this occurs too late in the disease pathology to restore blood pressure [8]. Additionally, genetically augmenting the renal lymphatic network can prevent HTN by trafficking pro-inflammatory immune cells out of the kidney [8].

Angiotensin II (Ang II), salt, and asymmetric dimethylarginine (ADMA) are known to induce HTN in mice and are elevated in humans with HTN. These HTN stimuli induce vasoconstriction, immune system activation, inflammation, and sodium retention [9]. Machnik et al. reported that a high salt diet in mice induced sodium storage in the skin, which caused macrophages to release lymphangiogenic growth factors [10]. When vascular endothelial growth factor C (VEGF-C) was inhibited, mice became hypertensive [10].

Our laboratory has reported previously that Ang II induces immune cells to secrete VEGF-C, which leads to renal lymphangiogenesis [11]. However, there are conflicting reports regarding whether Ang II acts directly on LECs [11,12]. The nature by which HTN stimuli induce lymphangiogenesis is unknown. We hypothesized that increased lymphangiogenesis is primarily driven by the direct effects of HTN stimuli on LECs, rather than indirect effects, such as secreted immune cell factors.

## 2. Materials and Methods

Study approval: All procedures performed in mice were approved by the Texas A&M University IACUC in accordance with the NIH Guide for the Care and Use and Care of Laboratory Animals.

Statistics: Data are presented as mean + SEM or mean ± SEM. Statistical analyses were performed using SigmaPlot 10 (Systat Software, Inc, San Jose, CA, USA). Two-tailed Student’s *t* test was used for analysis of data between two groups. One-way ANOVA followed by a Tukey post-hoc analysis was used for testing significant differences between more than two groups. Significance was denoted at *p* < 0.05.

Culture of Human Lymphatic Endothelial Cells (hLECs): hLECs used were purchased from Promocell (Cat. No. C-12216/C-12217, Heidelberg, Germany). hLECs were cultured and passaged (p4–p5) in complete EGM-MV2 media until they reached 75% confluency. Then, cells were treated with HTN stimuli, either 1 μM saline control, 1 μM Ang II (normal = 5 ηM), 190 mM salt (normal = 110 mM), or 100 μM ADMA (normal = 10 μM) for 24 h. The cells were routinely verified for the LEC markers lymphatic vessel endothelial hyaluronan receptor 1 (Lyve1), prospero-related homeobox 1 (Prox1), and podoplanin (Pdpn). Total RNA was extracted, and cDNA was reverse transcribed using 400 ng total RNA, as described previously [8]. qPCR was conducted to determine the gene expression of the proliferation markers Pcna and Ki67 using ubiquitin (Ubiq) as a reference gene. Additional hLECs underwent treatment as described above and were seeded onto 3-D collagen matrix gels to begin the invasion assay.

Invasion Assay: 3-D collagen matrices were made with Type I Collagen and seeded into 96-well plates as described previously [13]. Following 24 h treatment with HTN stimuli or saline, hLECs were seeded onto the 3-D collagen gels and allowed to invade for 24 h. Collagen gels included VEGF-C, fibronectin, and sphingosine 1-phosphate (S1P) to enhance invasion. VEGF-C, VEGF-A, basic fibroblast growth factor (bFGF), and hepatocyte growth factor (hGF) were in the media to stimulate lymphangiogenesis, as these growth factors have been optimized previously for hLECs [14]. Gels were fixed and stained with toluidine blue. The average number of sprouts, average invasion distance, and average lumen diameter were quantified. The number of cells per lumen structure was also determined. Average number of lumen-forming structures was quantified manually per 0.25 mm^2^ field using an eyepiece fitted with a 10 × 10 ocular grid. Manual counting was performed by two blinded investigators.

Culture of Mouse Lymphatic Endothelial Cells (mLECs): mLECs were purchased from Cell Biologics (Cat. No. C57-6064L, Chicago, IL, USA). mLECs were cultured and passaged (p4–p6) until they reached 85–90% confluency. The cells were plated into a 12-well plate and treated with HTN stimuli for 24 h once they reached 70% confluency. Cells were verified for the LEC markers Lyve1, Prox1, and Pdpn. qPCR was used to determine the gene expression of the proliferation markers Pcna and Ki67 using Ubiq as a reference gene.

Mesometrium Culture Model: Murine mesometrium is rich in LVs and the tissue is thin and transparent enough for easy visualization of vessel growth. Suarez-Martinez et al. reported that microvascular networks found in the mouse mesometrium can be used to examine and visualize angiogenesis [15]. We modified this model to visualize lymphangiogenesis. Mesometrium tissue explants were taken from six-month-old female Prox-1-tdTomato mice (*n* = 3) following isoflurane anesthesia and exsanguination euthanization. They were then cultured in DMEM supplemented with 20% fetal bovine serum to induce lymphatic sprouting. Mesometrial tissue explants were treated with either HTN stimuli directly or with HTN conditioned media for 8 days. Conditioned media was made by treating murine splenocytes for 24 h with the same HTN stimuli. Both media and treatments were replenished daily. Time lapse images were taken of multiple vessels per tissue explant once every 24 h using confocal microscopy. ImageJ was used to quantify sprout number and sprout length.

## 3. Results

Treatment with HTN stimuli did not significantly impact Lyve1, Prox1, or Pdpn gene expression (data not shown). There were also no significant changes in the gene expression of the proliferation markers PCNA or Ki-67 (Figure 1). Thus, HTN stimuli was found to not directly increase hLEC proliferation.

In the 3-D collagen invasion assay, the cells treated with HTN stimuli exhibited a significant increase in the number of lumen-forming structures and the overall invasion distance compared to the saline-treated control cells (Figure 2A–C). However, treatment with HTN stimuli was associated with a significant decrease in the average lumen diameter and the number of cells per invading structure (Figure 2A,D,E). In all HTN stimuli treatment groups, there was increased sprout formation, and notably, these sprouts were longer and thinner than those observed in the control vessels.

In mLECs, there were no significant changes in gene expression for lymphatic (data not shown) or proliferation markers (Figure 3) after 24 h treatments compared to controls, indicating that HTN stimuli did not directly induce mLEC proliferation.

In the mesometrial vessels, treatment with HTN stimuli significantly decreased mesometrial lymphatic sprout length and sprout number (control sprout length in pixels by ImageJ analysis: 34.0 ± 2.6, Ang II: 3.7 ± 2.6, salt: 2.67 ± 2.18, ADMA: 9.06 ± 5.12, all *p* < 0.05; control sprout number: 7 ± 3, Ang II: 0 ± 0, salt: 0 ± 0, ADMA: 1 ± 1, all *p* < 0.05; Figure 4). Conditioned media treatment normalized sprout length and sprout number by Day 8 for Ang II and ADMA but not for salt (Figure 4).

In the pro-lymphangiogenic saline control, an increase in lymphangiogenesis was observed from day 1 to day 8, in the absence of conditioned medium (Figure 4). However, a decrease in lymphangiogenesis was observed in LVs treated with HTN stimuli alone over the same time (Figure 4). In the presence of conditioned media, there was an increase in lymphangiogenesis for the saline control, but less than the saline control alone (Figure 5). Contrary to the HTN stimuli alone, there was significantly increased lymphangiogenesis in LVs treated with HTN stimuli in the presence of conditioned media from HTN-stimuli-treated splenocytes (Figure 5).

## 4. Discussion

The major findings of the current study are that HTN stimuli indirectly prevent a decrease in lymphangiogenesis through the presence of immune-cell-conditioned media and growth factors but do not directly induce LEC proliferation. Proliferation differs from lymphangiogenesis in the sense that proliferation involves the growth and increase in new cells, while lymphangiogenesis is the formation of new LVs. The interstitial fluid that is in direct contact with LECs contains similar or even higher levels of HTN stimuli compared to the blood [16]. Previously, it was unknown whether HTN stimuli directly induce LEC proliferation and/or lymphangiogenesis. Our results indicate that HTN stimuli do not directly induce LEC proliferation. Interstitial accumulation of activated immune cells at inflammatory sites causes them to secrete growth factors and regulatory factors that maintain homeostasis [17]. The junctions between LECs are specialized to regulate lymphatic fluid and immune cells to enter the LVs [18]. Thus, it is possible that HTN stimuli need the presence of high levels of growth factors to induce lymphangiogenesis.

A previous report demonstrated that Ang II was able to induce lymphangiogenesis and proliferation directly as measured by 5-bromo-2-deoxyuridine (BrdU) incorporation assays [12]. This was in contrast to our original report that Ang II did not have a direct effect on LEC proliferation [11]. Their results showed that mLEC proliferation increased in a time-dependent manner when treated with Ang II via the Ang II receptor AT1R. They found that Ang II treated LECs upregulated LYVE-1, Prox1, VEGF-C, and VEGFR-3 and altered Kdr, c-Jun, Stc1, Sparc, and Ets1 [12]. The methods of proliferation measurement used in the two reports differ, which may also explain the different results. The previously mentioned report used a visual representation using the BrdU incorporation assay and relative quantification [12]. In the present study, we used qPCR to measure the gene expression of two proliferation markers, PCNA and Ki-67. The previous report also treated the cells with a lower concentration of 0.5 μM Ang II compared to the 1 μM of Ang II that we used to treat the cells. These, as well as other potential factors, may explain the observed differences.

In this study, all HTN stimuli induced an increase in the number of lumen-forming structures and the invasion distance in 3-D collagen matrices, along with a decrease in lumen diameter and the number of cells per invading structure. These data demonstrate that HTN stimuli in the presence of growth factors caused LV sprouting, but not the proliferation of cells. The resulting structures had lumens and went deeper into the 3-D collagen matrix, although the lumens had decreased width and there were fewer cells per structure. Therefore, HTN stimuli interact with various growth factors in the media and extracellular matrix to cause the formation of longer, thinner vessels. In the context of HTN, this could indicate that LECs are forming new vessels that extend further into the interstitial space to potentially clear more immune cells and help resolve inflammation. These results imply that growth factors are necessary for HTN stimuli-induced lymphangiogenesis.

Angiopoietins have been shown to have low mitogenic or proliferative potential for endothelial cells; however, both angiopoietin 1 and angiopoietin 2 increase endothelial cell migration and sprouting angiogenesis [19,20,21]. Previously, we have also demonstrated that while stimulation with the inflammatory agent lipopolysaccharide induced LEC proliferation, treatment with lipopolysaccharide decreased lymphatic sprouting and invasion [14]. This is in keeping with the findings of this study showing opposite correlation between proliferation and invasion of LECs. However, further experiments are warranted to explore the detailed mechanisms at play in this regard. Despite a lack of significant change in hLEC gene expression for proliferative markers, there was significant growth of LVs into the 3-D collagen gels. It is known that a variety of growth factors can induce cell proliferation, but no proliferation was seen in the hLECs and mLECs when treated with HTN stimuli.

The results from the mLEC experiments supported the findings from the hLEC experiments. Again, HTN stimuli alone did not affect the proliferation of mLECs. In mesometrium LVs, all treatment groups were in the presence of 20% FBS to promote lymphangiogenesis. HTN stimuli alone inhibited lymphangiogenesis. However, when treated with HTN stimuli-conditioned media, the decrease in lymphangiogenesis was attenuated, indicating that immune cell secretions were necessary for inducing lymphangiogenesis. These results corroborate those from the hLEC 3-D collagen invasion assay in that immune cell-secreted growth factors were required for lymphangiogenesis. It should be noted that while the mesometrium LVs were treated with HTN conditioned media, the invasion assay utilizing hLECs were treated with HTN stimuli directly and allowed to invade in the presence of growth factors in the 3-D collagen gel, whereas direct HTN treatment inhibited sprouting in the mesometrium LVs. These differences in direct HTN treatment could be due to the tissue types, human and mouse. The invasion assay had hypertensive-stimuli-treated cells without hypertensive stimuli in the media or gel. The mesometrium saline treatment is a positive control from the 20% FBS culture media, which promoted lymphangiogenesis on its own. A variety of factors could have influenced this result. Further analysis would be needed to determine the cause of this difference. The invasion assay gel and media include various growth factors that enhance lymphangiogenesis, including VEGF-C, HGF, S1P, and bFGF, which are also secreted by immune cells. Activated immune cells, notably macrophages, are also known to secrete VEGF-C and VEGF-D and induce lymphangiogenesis [22]. Therefore, the growth factors in the invasion assay and the factors in the HTN-stimuli-conditioned media are likely very similar. Further testing of the HTN-conditioned media can confirm this. Nonetheless, these secreted immune cell factors are necessary to promote lymphangiogenesis in both assays.

It is unclear which type of immune cell is responsible for the observed results, but macrophages have been known to cause lymphangiogenesis [10]. A variety of immune cells were used in the current study to mirror the microenvironment of lymphatics since there is an abundant amount of various immune cells in the interstitial space during inflammation in HTN [23]. According to Miltenyi Biotec and others, mouse splenocytes typically consist of T cells (21–25% of total cells), B cells (44–58%), monocytes (3.5–5%), granulocytes (1–2%), dendritic cells (1–3%), natural killer cells (1–2%), and macrophages (1–2%). Studies are currently underway to systematically determine the contribution of each immune cell type to the HTN-stimuli-induced secretion of factors that induce lymphangiogenesis.

A future direction would be to analyze the composition of the conditioned media and whether some of those factors are similar to the growth factors used in the invasion assay. Another topic of interest would be to investigate how hLECs treated with HTN conditioned media differ from the HTN stimuli alone in the invasion assay. The invasion assay could include not only HTN-stimuli-treated cells but also HTN stimuli within the media and collagen gel. Investigating the reason for the difference in results between the HTN stimuli alone in the invasion assay and the mesometrial LVs could be performed. The hLEC and mLEC PCR Figure 1 and Figure 3 have some groups that look like they could be significantly different but are not, so it would be valuable to repeat the cultures with HTN stimuli to obtain a larger sample size. Nonetheless, the lack of LEC proliferation by HTN stimuli is consistent with an increased invasion ability, as proliferating cells would spread across the top of the 3-D collagen matrix instead of invading and forming lumen-containing structures.

There are various ways to treat and prevent HTN, including lifestyle changes such as eating healthy and exercising, along with medications that inhibit some HTN stimuli [24]. Despite this, in the United States, 30% of hypertensive adults are unaware that they have HTN, 40% are not receiving treatment, and only 30% have their blood pressure under control [25]. With HTN being a major public health problem, it is critical to develop more therapeutic options to control, treat, and prevent the disease. Currently, there are no FDA-approved therapeutics to our knowledge designed to target lymphatics to combat or control HTN. By determining the mechanism that mediates HTN-induced lymphangiogenesis, therapeutics may be developed that have direct clinical applications in the treatment of HTN. Lowering the rate of HTN lowers the incidence of HTN-related deaths, thus increasing the overall health of the population.

In conclusion, the main results of this study include that HTN stimuli have no direct effect on hLEC or mLEC proliferation but promote endothelial morphogenesis in the invasion assay (Figure 6). In the mesometrial LVs, HTN stimuli alone inhibited lymphangiogenesis, but that decrease was attenuated in the presence of conditioned media from HTN-stimuli-treated immune cells.

## Figures and Tables

**Figure 1 cells-11-02139-f001:**
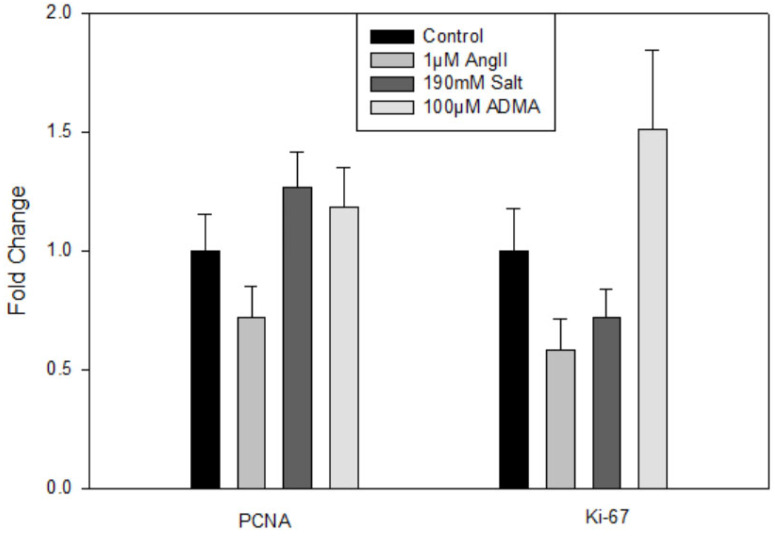
Treatment with HTN stimuli does not increase hLEC proliferation. mRNA expression of PCNA and Ki-67 in hLECs treated with saline, Ang II, salt, or ADMA. Results are expressed as mean + SEM (*n* = 7–12 independent experiments) and statistical analyses were performed with one-way ANOVA.

**Figure 2 cells-11-02139-f002:**
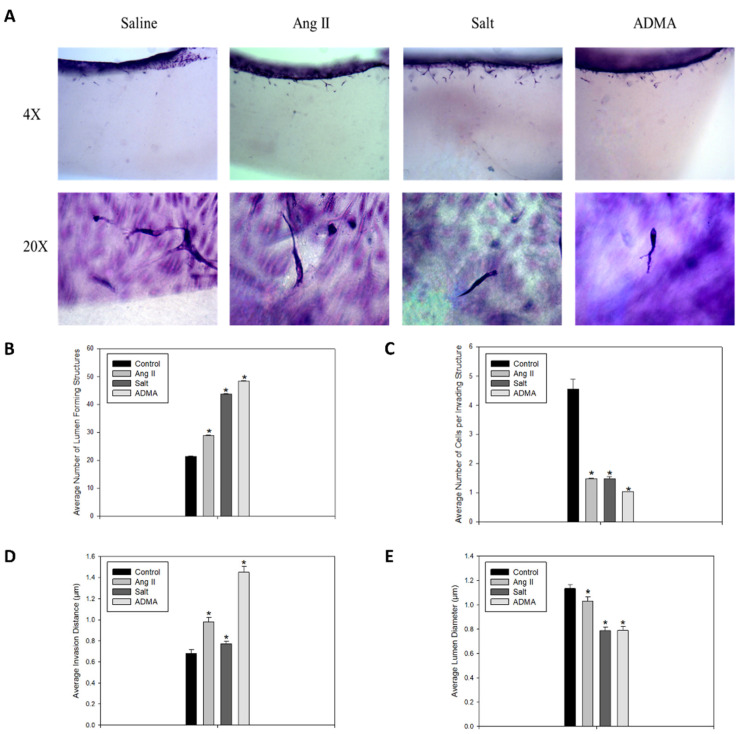
Treatment with HTN stimuli induced the formation of longer, thinner lymphatic vessels in hLECs. These figures show this with the inverse relationship between figures (**B**,**D**) and (**C**,**E**). Following the 24 h invasion assay, gels stained with toluidine blue were cut into 4–5 slices. Gels were imaged on their sides to allow for visualization of both the monolayer and the gel. (**A**) Images at 4× (invasion distance) and 20× (lumen diameter) were taken of LV sprouts in hLECs treated with saline, Ang II, salt, or ADMA. (**B**) The average number of lumen-forming structures (*n* = 150 or more), (**C**) average number of cells per invading structure (*n* = 187 or more), (**D**) average invasion distance (*n* = 475 or more), and (**E**) average lumen diameter (*n* = 150 or more) were quantified for all groups. Nuclei within sprouting vessels were counter-stained with DAPI to allow for quantification of the average number of cells per invading structure in each group. Invasion distance is defined as the minimum distance from the monolayer to the tip of the invading structure. The lumen of the vessel is defined as the clear space free of nuclei within a continuous membrane, which was visualized with the toluidine blue stain. ImageJ was used to measure the maximum width of the clear spaces. Results are expressed as mean + SEM, and statistical analyses were performed with a one-way ANOVA. * *p* < 0.05 vs. saline-treated cells.

**Figure 3 cells-11-02139-f003:**
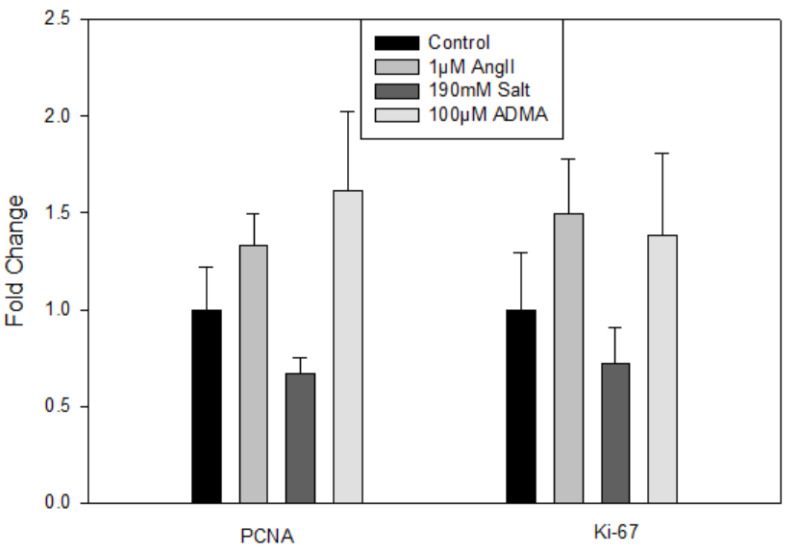
Treatment with HTN stimuli does not increase mLEC proliferation. mRNA expression of PCNA and Ki-67 in mLECs treated with saline, Ang II, salt, or ADMA. Results are expressed as mean + SEM (*n* = 7–13 independent experiments), and statistical analyses were performed with a one-way ANOVA.

**Figure 4 cells-11-02139-f004:**
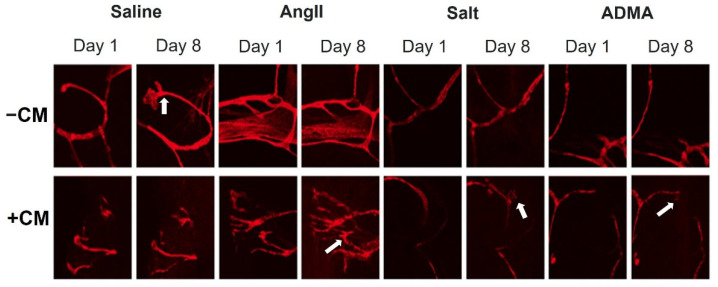
Treatment with HTN-stimuli-conditioned media rescues the decrease in average sprout length and average sprout number caused by treatment with HTN stimuli alone in murine mesometrial LVs. Murine mesometrial LVs treated with HTN stimuli with and without conditioned media from Day 1 to Day 8. Images were taken at 20× magnification.

**Figure 5 cells-11-02139-f005:**
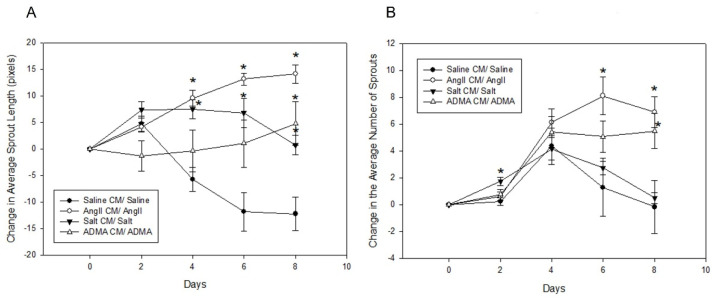
Treatment with HTN-stimuli-conditioned media attenuated the decrease in sprout length and sprout number (aside from salt) caused by direct treatment with HTN stimuli in murine mesometrium LVs. These delta figures combine the direct HTN stimuli treatment groups and the HTN-stimuli-conditioned media treatment groups in order to better visualize the attenuated decrease when compared to the saline control. (**A**) Average sprout length and (**B**) average sprout number, as rationalized in our previous publication [14], in murine mesometrium explants treated with HTN stimuli in the presence and absence of conditioned media. Results are expressed as mean ± SEM, and statistical analyses were performed with a one-way ANOVA. * *p* < 0.05 vs. saline-treated LVs.

**Figure 6 cells-11-02139-f006:**
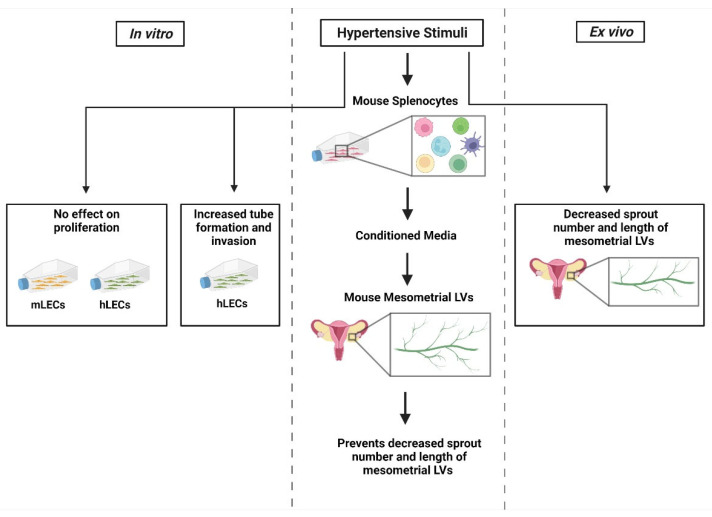
Summary of the effects that HTN stimuli and HTN-stimuli-conditioned media have on mouse and human LECs, as well as mouse mesometrial lymphatic vessels (LVs). HTN stimuli had no direct effect on LEC proliferation under normal conditions but did influence lymphatic vessel formation under pro-lymphangiogenic conditions.

## Data Availability

All data are presented in this paper.

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
