# Peer review of "Hypertensive Stimuli Indirectly Stimulate Lymphangiogenesis through Immune Cell Secreted Factors"

_cells, 2022, doi:10.3390/cells11142139_

Round 1
Reviewer 1 Report
I appreciate the focus of the manuscript. The lymphangiogenesis-immune cell-hypertension axis is novel, understudied, important and interesting. Because of this I am enthusiastic about the manuscript. I found the writing and structure to be straightforward and easy to follow. Strengths include multiple assays, multiple stimuli, and the inclusion of HTN stimulated immune cell media. Please see below for specific suggestions to improve the presentation/discussion of the results.
1. A major result is that “Conditioned media from HTN stimuli-treated splenocytes significantly increased the sprout number (aside from salt) and sprout length of mouse mesometrial LVs.” This statement seems to be different that those in the results and discussion indicating that a conditioned media “attenuated” or “rescued” the responses. I suggest ensuring statements about the results are consistent. Did HTN stimuli alone cause lymphangiogenesis. This is important because the conclusion states that – “In mesometrium LVs, HTN stimuli alone inhibited lymphangiogenesis. However, when treated with HTN stimuli-conditioned media, the decrease in lymphangiogenesis was attenuated, indicating that immune cell secretions were necessary for inducing lymphangiogenesis.” So, did the conditioned media cause lymphangiogenesis or just rescue a decrease. Adding text to step through your interpretation will help guide the reader. Also, I suggest adding rationale for your analysis metric for Figure 5.
2. A critical result seems to be that there is something in the conditioned media that is different than the HTN stimuli. Based on the results (though different assays) HTN stimuli and conditioned media cause lymphangiogenesis. I suggest adding to the Discussion to elaborate on your interpretation. One take is that the HTN stimuli cause lymphangiogenis in some assays but not the sprouting assay for Figure 5. I am OK with the potential differences given the novelty of the data, but do think these details should be recognized.
3. A follow up question is – What is in the conditioned media? While I do not suggest additional experiments, I do suggest speculating what the follow up studies might be.
4. I suggest defining the cell populations within splenocytes. I am not sure if you did this.
5. Figure 2: Provide rationale for concentrations.
6. Figure 1/3: I appreciate that there are not significant differences, but some of the bars look different. I suggest adding text to the Discussion to emphasize how the data supports your interpretation.
7. I suggest adding scale bars and annotations (arrows) to the images. For example, I am not sure what I am looking at in the bottom row of images in Figure 2. Is the key detail the diameter or the number of segments? I assume diameter because I only see 1 structure per image. Also, please explain what I should be looking at in the Figure 4 images.
8. “Conclusions: These data indicate that HTN stimuli indirectly increase lymphangiogenesis through secreted factors from HTN stimuli-treated immune cells.” Again – this statement warrants a more guided presentation of your interpretation of the data.
9. “The major findings of the current study are that HTN stimuli indirectly cause an increase in lymphangiogenesis through the presence of immune cell-conditioned media and growth factors, but do not directly induce LEC proliferation.” I suggest editing this statement because your initial HTN stimuli data does seem to support a lymphangiogenic effect. Also, I suggest editing the text in other locations of the manuscript to recognize that the responses were not consistent across the 3 HTN stimuli. Figure 5 suggests the Salt CM effect is different. Figure 2 suggests that not all 3 effects are the same.
Author Response
I appreciate the focus of the manuscript. The lymphangiogenesis-immune cell-hypertension axis is novel, understudied, important and interesting. Because of this I am enthusiastic about the manuscript. I found the writing and structure to be straightforward and easy to follow. Strengths include multiple assays, multiple stimuli, and the inclusion of HTN stimulated immune cell media. Please see below for specific suggestions to improve the presentation/discussion of the results.
Thank you for your help and support!
- A major result is that “Conditioned media from HTN stimuli-treated splenocytes significantly increased the sprout number (aside from salt) and sprout length of mouse mesometrial LVs.” This statement seems to be different than those in the results and discussion indicating that a conditioned media “attenuated” or “rescued” the responses. I suggest ensuring statements about the results are consistent. Did HTN stimuli alone cause lymphangiogenesis. This is important because the conclusion states that – “In mesometrium LVs, HTN stimuli alone inhibited lymphangiogenesis. However, when treated with HTN stimuli-conditioned media, the decrease in lymphangiogenesis was attenuated, indicating that immune cell secretions were necessary for inducing lymphangiogenesis.” So, did the conditioned media cause lymphangiogenesis or just rescue a decrease. Adding text to step through your interpretation will help guide the reader. Also, I suggest adding rationale for your analysis metric for Figure 5.
We agree with your assessment regarding the statement of the findings and have made the following revisions throughout the manuscript to state the results more accurately.
In abstract, line 23, corrected from “Conditioned media from HTN stimuli-treated splenocytes significantly increased the sprout number (aside from salt) and sprout length of mouse mesometrial LVs.” to “Conditioned media from HTN stimuli-treated splenocytes significantly attenuated the decrease in sprout number (aside from salt) and sprout length of mouse mesometrial LVs that is found in the HTN stimuli alone.”
Line 245, corrected from “In mesometrium LVs, HTN stimuli alone inhibited lymphangiogenesis. However, when treated with HTN stimuli-conditioned media, the decrease in lymphangiogenesis was attenuated, indicating that immune cell secretions were necessary for inducing lymphangiogenesis.” to “In mesometrium LVs, all treatment groups were in the presence of 20% FBS to promote lymphangiogenesis. HTN stimuli alone inhibited lymphangiogenesis. However, when treated with HTN stimuli-conditioned media, the decrease in lymphangiogenesis was attenuated, indicating that immune cell secretions were necessary for inducing lymphangiogenesis.”
Line 183, corrected from “sprout length and sprout number caused by direct treatment with HTN stimuli in…” to “sprout length and sprout number (aside from salt) caused by direct treatment with HTN stimuli in…”.
Line 184, we inserted this sentence “These delta figures combine the direct HTN stimuli treatment groups and the HTN stimuli conditioned media treatment groups in order to better visualize the attenuated decrease when compared to the saline control.”
Line 167, corrected from “In the saline control, an increase in lymphangiogenesis was observed from day 1 to day…” to “In the pro-lymphangiogenic saline control, an increase in lymphangiogenesis was observed from day 1 to day…”
Line 188, We have referenced our previous paper regarding the analysis metric for figure 5 in the figure legend.
- A critical result seems to be that there is something in the conditioned media that is different than the HTN stimuli. Based on the results (though different assays) HTN stimuli and conditioned media cause lymphangiogenesis. I suggest adding to the Discussion to elaborate on your interpretation. One take is that the HTN stimuli cause lymphangiogenesis in some assays but not the sprouting assay for Figure 5. I am OK with the potential differences given the novelty of the data but do think these details should be recognized.
Revisions:
Line 251, “It should be noted…” refers to this difference in results between assays.
Line 254, we inserted this sentence “Whereas direct HTN treatment inhibited sprouting in the mesometrium LVs. These differences in direct HTN treatment could be due to the tissue types, human and mouse. The invasion assay had hypertensive stimuli treated cells without hypertensive stimuli in the media or gel. The mesometrium saline treatment is a positive control from the 20% FBS culture media which promoted lymphangiogenesis on its own. A variety of factors could have influenced this result. Further analysis would be needed to determine the cause of this difference.”
- A follow up question is – What is in the conditioned media? While I do not suggest additional experiments, I do suggest speculating what the follow up studies might be.
Line 279, we inserted a new paragraph “A future direction would be to analyze the composition of the conditioned media and whether some of those factors are similar to the growth factors used in the invasion assay. Another topic of interest would be to investigate how hLECs treated with HTN conditioned media differ from the HTN stimuli alone in the invasion assay. The invasion assay could include not only HTN stimuli treated cells, but also HTN stimuli within the media and collagen gel. Investigating the reason for the difference in results between the HTN stimuli alone in the invasion assay and the mesometrial LVs. The hLEC and mLEC PCR figures 1 and 3 have some groups that look like they could be significantly different but are not, so it would be valuable to repeat the cultures with HTN stimuli to get a larger sample size for more confident statistical accuracy.”
- I suggest defining the cell populations within splenocytes. I am not sure if you did this.
Line 273 we inserted this sentence “According to Miltenyi Biotec and others, mouse splenocytes typically consist of T cells (21-25% of total cells), B cells (44-58%), monocytes (3.5-5%), granulocytes (1-2%), dendritic cells (1-3%), natural killer cells (1-2%), and macrophages (1-2%).”
- Figure 2: Provide rationale for concentrations.
Line 78, we added normal values for the HTN stimuli “Then, cells were treated with HTN stimuli, either 1 μM saline control, 1 μM Ang II (normal=5 ηM), 190 mM salt (normal=110 mM), or 100 μM ADMA (normal=10 μM) for 24 hours.”
- Figure 1/3: I appreciate that there are not significant differences, but some of the bars look different. I suggest adding text to the Discussion to emphasize how the data supports your interpretation.
Line 288, we inserted the following “Nonetheless, the lack of LEC proliferation by HTN stimuli is consistent with an increased invasion ability as proliferating cells would spread across the top of the 3-D collagen matrix instead of invading and forming lumen containing structures.”
- I suggest adding scale bars and annotations (arrows) to the images. For example, I am not sure what I am looking at in the bottom row of images in Figure 2. Is the key detail the diameter or the number of segments? I assume diameter because I only see 1 structure per image. Also, please explain what I should be looking at in the Figure 4 images.
Line 140, corrected from “(a) 4X and 20X images were taken of LV sprouts…” to “(a) 4X (invasion distance) and 20X (lumen diameter) images were taken of LV sprouts”. We have also added white arrows to the lymphatic sprouts in the images for Figure 4.”
- “Conclusions: These data indicate that HTN stimuli indirectly increase lymphangiogenesis through secreted factors from HTN stimuli-treated immune cells.” Again – this statement warrants a more guided presentation of your interpretation of the data.
This has been changed from “Conclusions: These data indicate that HTN stimuli indirectly increase lymphangiogenesis through secreted factors from HTN stimuli-treated immune cells.” to Line 25, this has been changed to “Conclusions: These data indicate that HTN stimuli indirectly prevent a decrease in lymphangiogenesis through secreted factors from HTN stimuli-treated immune cells.”
- “The major findings of the current study are that HTN stimuli indirectly cause an increase in lymphangiogenesis through the presence of immune cell-conditioned media and growth factors, but do not directly induce LEC proliferation.” I suggest editing this statement because your initial HTN stimuli data does seem to support a lymphangiogenic effect. Also, I suggest editing the text in other locations of the manuscript to recognize that the responses were not consistent across the 3 HTN stimuli. Figure 5 suggests the Salt CM effect is different. Figure 2 suggests that not all 3 effects are the same.
Line 192, the first sentence of the Discussion has been updated to be consistent with the findings as stated above.
For Figure 5, see correction referred to in comment #1.
For Figure 2, line 137, we inserted this sentence “These figures show this with the inverse relationship between figures B and D and figures C and E.”
Reviewer 2 Report
In this article, Wilcox et al report the data of experiments performed to investigate the relationship between hypertension and lymphangiogenesis. They found that stimulating hypertension by use of angiotensin II, salt, or ADMA have a direct effect on lymphangiogenesis in either a cell or an in vivo model (mouse mesometrium). Remarkably, the various stimuli appeared to influence the number of lumen-forming structures and invasion distance rather than cell proliferation, highlighting the presence of factors secreted by immune cells.
The study appears well done and interesting; however, a summarizing figure or scheme would be useful to understand the whole picture.
The lack of influence of HTN stimuli on cell proliferation is described in Figures 1 and 3. However, performing more experiments (n=5 and 4, respectively, now) may let some statistical significance emerge. Report the values for one-way ANOVA followed by less conservative post-hoc tests like Tukey’s. This is one of the few cases where absence of statistical significance is the target, please use adequate statistical tools to demonstrate it.
A few-lines conclusion summarizing the main results of the study is recommended.
Minor:
Line 69: “mean + or ± SEM”, perhaps something is missing.
Are the Authors sure that Human Lymphatic Endothelial Cells are good representative of lymphatic vessels? Perhaps a reference is needed in support of this. Is this reference 8?
Line 103: Specify the treatment of the animals (age, anesthesia, etc).
Line 203, rearrange the sentence in order to disnguish the molecules used for treatment of LECs from those that were upregulated.
Author Response
In this article, Wilcox et al report the data of experiments performed to investigate the relationship between hypertension and lymphangiogenesis. They found that stimulating hypertension by use of angiotensin II, salt, or ADMA have a direct effect on lymphangiogenesis in either a cell or an in vivo model (mouse mesometrium). Remarkably, the various stimuli appeared to influence the number of lumen-forming structures and invasion distance rather than cell proliferation, highlighting the presence of factors secreted by immune cells.
- The study appears well done and interesting; however, a summarizing figure or scheme would be useful to understand the whole picture.
We have added a summary figure to the end of the discussion (Figure 6).
- The lack of influence of HTN stimuli on cell proliferation is described in Figures 1 and 3. However, performing more experiments (n=5 and 4, respectively, now) may let some statistical significance emerge. Report the values for one-way ANOVA followed by less conservative post-hoc tests like Tukey’s. This is one of the few cases where absence of statistical significance is the target, please use adequate statistical tools to demonstrate it.
Unfortunately, we are not able to increase the sample size given the 5-day deadline to return the revised manuscript. However, we have added this as a limitation in the Discussion (Line 285). We re-analyzed the data in Figures 1 and 3 with a one-way ANOVA followed by a Tukey post-hoc test and there were still no statistical differences. This has been updated in the methods (Line 72).
- A few-lines conclusion summarizing the main results of the study is recommended.
Insert into end of discussion.
Line 303, we inserted the following “In conclusion, the main results of this study include that HTN stimuli have no direct effect on hLEC or mLEC proliferation but promote endothelial morphogenesis in the invasion assay (Figure 6). In the mesometrial LVs, HTN stimuli alone inhibited lymphangiogenesis but that decrease was attenuated in the presence of conditioned media from HTN stimuli-treated immune cells.”
- Line 69: “mean + or ± SEM”, perhaps something is missing.
Line 70 corrected from “mean + or ± SEM” to “mean + SEM or mean ± SEM.”
- Are the Authors sure that Human Lymphatic Endothelial Cells are good representative of lymphatic vessels? Perhaps a reference is needed in support of this. Is this reference 8?
Reference 14 characterizing LEC 3-D collagen gel invasion and formation of lymphatic vessels supports this.
- Line 103: Specify the treatment of the animals (age, anesthesia, etc).
Line 109 corrected from “Mesometrium tissue explants were taken from female Prox-1-tdTomato mice” to “Mesometrium tissue explants were taken from six-month-old, female Prox-1-tdTomato mice (n=3) following isoflurane anesthesia and exsanguination euthanization.”
- Line 203, rearrange the sentence in order to distinguish the molecules used for treatment of LECs from those that were upregulated.
Line 211 corrected from “…found that when LECs were treated with Ang II, LYVE-1, Prox1, VEGF-C, and VEGFR-3 were upregulated and Kdr, c-Jun, Stc1, Sparc, and Ets1 were altered.[12]” to “found that AngII treated LECs upregulated LYVE-1, Prox1, VEGF-C, and VEGFR-3 and altered Kdr, c-Jun, Stc1, Sparc, and Ets1.[12]” Thank you for the helpful comments!